# Olmesartan Attenuates Kidney Fibrosis in a Murine Model of Alport Syndrome by Suppressing Tubular Expression of TGFβ

**DOI:** 10.3390/ijms20153843

**Published:** 2019-08-06

**Authors:** Sang Heon Suh, Hong Sang Choi, Chang Seong Kim, In Jin Kim, Seong Kwon Ma, James W. Scholey, Soo Wan Kim, Eun Hui Bae

**Affiliations:** 1Department of Internal Medicine, Chonnam National University Hospital, Gwangju 61469, Korea; 2Department of Medicine and Institute of Medical Science, University of Toronto, Toronto, NO M5S, Canada

**Keywords:** Alport syndrome, olmesartan, renin-angiotensin system, transforming growth factor β, angiotensin-converting enzyme 2

## Abstract

Despite the wide use of angiotensin II receptor blockers in the treatment of Alport syndrome (AS), the mechanism as to how angiotensin II receptor blockers prevent interstitial fibrosis remains unclear. Here, we report that treatment of olmesartan effectively targets the feedback loop between the renin–angiotensin system (RAS) and transforming growth factor β (TGFβ) signals in tubular epithelial cells and preserves renal angiotensin-converting enzyme 2 (ACE2) expression in the kidney of *Col4a3^–/–^* mice, a murine model of experimental AS. Morphology analyses revealed amelioration of kidney fibrosis in *Col4a3^–/–^* mice by olmesartan treatment. Upregulation of TGFβ and activation of its downstream in *Col4a3^–/–^* mice were attenuated by olmesartan in *Col4a3^–/–^* mice. Intriguingly, TGFβ expression was preferentially upregulated in damaged tubular epithelial cells in *Col4a3^–/–^* mice. Concurrent upregulation of TNFα-converting enzyme and downregulation of ACE2 suggested RAS activation in *Col4a3^–/–^* mice, which was prevented by olmesartan. Mechanistically, olmesartan suppressed TGFβ-induced RAS activation in tubular epithelial cells in vitro. Collectively, we concluded that olmesartan effectively suppresses the progression of tubulointerstitial fibrosis in AS by interrupting RAS-TGFβ feedback loop to counterbalance intrarenal RAS activation.

## 1. Introduction

Alport syndrome (AS) is a hereditary type IV collagen disease caused by mutations in *COL4A3/4/5*, which encode α3/α4/α5 type IV collagen chains [1,2] with defects in postnatal maturation of glomerular basement membrane (GBM) [3,4]. Experimentally, *Col4a3*-deficient (*Col4a3^–/–^*) mice has been an animal model human AS [5], which develop progressive glomerulonephritis as well as tubulointerstitial scarring and die with end-stage renal disease at an age of approximately 10 weeks. Although glomeruli are the primary site of pathogenesis, tubulointerstitial fibrosis is both a key feature of disease progression and a therapeutic target of AS. Pre-emptive treatment with an angiotensin-converting enzyme inhibitor (ACEi) or angiotensin II receptor blocker (ARB) suppressed cytokine and collagen production and consequently attenuated inflammation and tubulointerstitial fibrosis in *Col4a3^–/–^* mice despite the persistent ultrastructural defect in GBM [6,7]. Ultrastructural analyses in those studies demonstrated that treatment with ramipril slightly improved foot process effacement, glomerular hypertrophy, and accumulation of intracellular fibrillar collagens; however, regardless of ramipril treatment, the characteristic thickening and splitting of GBM was observed in *Col4a3^–/–^* mice. Those studies also reported an outstanding anti-fibrotic effect of an ACEi or ARB both in the glomeruli and tubulointerstitium of *Col4a3^–/–^* mice.

Accumulating evidences suggest the pathogenic role of renin–angiotensin system (RAS) in AS. Blockade of RAS with ACEis or ARBs has demonstrated the renoprotective effect in clinical AS patients as well as in animal models [6,7,8,9]. ACE2, originally identified as a human homology of ACE, generates angiotensin-(1-9) from Ang I or degrades Ang II to the angiotensin-(1-7) (Ang-(1-7)) [10], although the enzyme efficacy for Ang II is 400-fold greater for Ang I [11]. In consideration of the generation from Ang II by ACE2 and the function as a vasodilator, Ang-(1-7) has been expected to present counter-regulatory mechanisms against the activation of classical RAS pathway involving ACE and Ang II [12,13]. The organ-specific role of ACE2 has been recently reported in the context of the anatomical restriction of ACE2 expression to the heart and kidneys [10]. Loss of ACE2 is associated with age-dependent development of glomerulosclerosis and albuminuria [14] as well as exacerbation of diabetic kidney injury in mice [15]. Our group also recently reported the protective effect of recombinant ACE2 and Ang-(1-7) on kidney fibrosis in AS nephropathy [16] and unilateral ureteral obstruction [17] models, respectively. Despite these intensive studies, our current understandings on RAS and ACE2 pathophysiology in AS do not explain clearly how ACE2 expression and its activity are regulated during disease progression.

Olmesartan is an Ang II type 1 receptor (AT1R) antagonist that can inhibit Ang II-induced cardiac remodeling and heart failure by lowering blood pressure and AT1R blockade [18,19]. Beyond the classical action of AT1R blockade, upregulation of ACE2 in heart tissue by olmesartan administration has been reported in stroke-prone spontaneously hypertensive rats [20], experimental rodent models of post-myocardial infarction, and cardiac myosin-induced dilated cardiomyopathy [21,22]. Recently, olmesartan inhibited cardiac hypertrophy through enhancement of ACE2/Ang-(1-7)/Mas axis, independently of the effect on blood pressure control [23].

Taken together, we hypothesized that olmesartan might effectively attenuate kidney fibrosis in AS model by upregulation of intrarenal ACE2 expression. Here, we treated *Col4a3^–/–^* mice with olmesartan between four to seven weeks of age and revealed that olmesartan protects kidney from progressive fibrosis via preserving ACE2 expression in *Col4a3^–/–^* mice. We mechanistically defined the feedback loop between RAS and transforming growth factor β (TGFβ) signals in tubular epithelial cells, which is opposed by olmesartan treatment.

## 2. Results

### 2.1. Olmesartan Ameliorates Kidney Fibrosis in Col4a3^–/–^ Mice

To assess the effect of olmesartan treatment on the kidney function in *Col4a3^–/–^* mice, we measured urinary excretion of creatinine and albumin. In comparison with WT mice, urine creatinine and albumin concentration was significantly decreased and increased, respectively, in *Col4a3^–/–^* mice, despite the absence of significant differences in the body and kidney weight as well as the kidney-to-body weight ratio between the groups, implying the well-tolerated oral dosage of olmesartan in *Col4a3^–/–^* mice, as seen in Table 1. Accordingly, the urine albumin-to-creatinine ratio significantly increased in *Col4a3^–/–^* mice, as seen in Table 1, suggesting glomerular injury. Interestingly, neutrophil gelatinase-associated lipocalin level significantly increased in both serum and urine from *Col4a3^–/–^* mice at seven weeks of age, as seen in Table 1, implying tubulointerstitial damage. Olmesartan treatment abrogated increased urinary excretion albumin and elevated level of neutrophil gelatinase-associated lipocalin both in serum and urine, as seen in Table 1. To visualize the tissue morphology, a series of histochemical staining, such as hematoxylin and eosin, periodic acid Schiff, and Masson’s trichrome staining, was evaluated in the kidney of WT and *Col4a3^–/–^* mice, as seen in Figure 1a. Contrary to the glomeruli of WT mice, fibrotic crescent formation was packing the Bowman’s space of *Col4a3^–/–^* mice. With shrinkage of mesangium, capillary loops were significantly collapsed in *Col4a3^–/–^* mice. In the tubulointerstitial compartment, detachment of tubular epithelial cells from basement membranes were observed in *Col4a3^–/–^* mice, suggesting tubular necrosis, in which collagen deposits in interstitial spaces were also clearly observable. Olmesartan treatment remarkably prevented such abnormal findings of glomeruli and tubulointerstitium in the histochemical stainings. To characterize kidney fibrosis at the molecular level, fibrosis markers were compared among the group. Upregulation of α smooth muscle actin (αSMA) and fibronectin (FN) by immunoblotting, as seen in Figure 1b, and upregulation of FN and collagen type 1 (Col1) by quantitative polymerase chain reaction (qPCR), as seen in Figure 1c in the whole kidney lysate of *Col4a3^–/–^* mice was substantially aborted by olmesartan administration. Immunohistochemical staining revealed that Col1 and αSMA expression was upregulated both in glomeruli and peritubular interstitium of *Col4a3^–/–^* mice, as seen in Figure 1d,e. Such alterations in Col1 and αSMA expression of *Col4a3^–/–^* mice were significantly counteracted by olmesartan treatment. Taken together, these data suggest that olmesartan ameliorates kidney fibrosis in *Col4a3^–/–^* mice.

### 2.2. Preferential Upregulation of TGFβ in Tubulointerstitium in Col4a3^–/–^ Mice is Attenuated by Olmesartan

To determine the underlying signals relevant to tissue fibrosis in *Col4a3^–/–^* mice, we focused on *TGFβ* pathway, a key player in the progression of tubulointerstitial fibrosis [6,24]. Indeed, immunoblotting, as seen in Figure 2a, and qPCR, as seen in Figure 2b, in whole kidney lysate revealed upregulation of TGFβ in *Col4a3^–/–^* mice, which was significantly reduced by olmesartan treatment. Increased expression of SMAD4, a canonical downstream of TGFβ receptor, was also markedly attenuated by olmesartan treatment; however, phosphorylation of SMAD2/3 was not different among the groups, as seen in Figure 2a. To define the specific anatomical site or cellular component with TGFβ upregulation, immunohistochemistry was checked, as seen in Figure 2c and Appendix A. Intriguingly, the expression of TGFβ was differentially regulated in both the glomeruli and tubulointerstitial compartments. Although upregulation of TGFβ was not prominent in the glomeruli of *Col4a3^–/–^* mice, as seen in Appendix A, intense TGFβ signals were observed in tubulointerstitial compartment, especially in the epithelial cells of markedly dilated, atrophied tubules of *Col4a3^–/–^* mice when compared with the other two groups, as seen in Figure 2c. Collectively, our findings suggest TGFβ signaling is upregulated more preferentially in tubular epithelial cells than in glomerulus of *Col4a3^–/–^* mice at the time of sacrifice, which is efficiently attenuated by olmesartan treatment. 

### 2.3. Downregulation of ACE2 in Col4a3^–/–^ Mice Is Counteracted by Olmesartan

To determine the signals that switch on the *TGFβ* signaling and those that are blocked by olmesartan, we investigated the RAS as a candidate upstream pathway. Immunoblotting of whole kidney lysate, as seen in Figure 3, demonstrated that, when compared with the WT and olmesartan-treated group, tissue Ang II-III level was markedly increased in *Col4a3^–/–^* mice; however, tissue expression of ACE was not different among the groups. Instead, ACE2 and TACE was significantly downregulated and upregulated, respectively, in *Col4a3^–/–^* mice, which was prevented by olmesartan administration. The expression of Ang II receptors was also differentially regulated. Although AT1R was upregulated, Ang II type 2 receptor (AT2R) was downregulated in *Col4a3^–/–^* mice. These results collectively indicate the activation of RAS in *Col4a3^–/–^* mice, which is effectively counter-regulated by olmesartan treatment.

To explain the detailed mechanism for alterations in Ang II receptors, ACE2, and TACE, we examined mitogen-activated protein kinases (MAPKs), which has been suggested to relay the signal between AT1R and TACE [25]. Immunoblotting of whole kidney lysate, as seen in Figure 4, showed that phosphorylation of P38, but not ERK or JNK, was enhanced in *Col4a3^–/–^* mice. Intriguingly, olmesartan blocked phosphorylation of P38 in *Col4a3^–/–^* mice. Accordingly, it was suggested that both activation of classical RAS pathway and downregulation of ACE2 in *Col4a3^–/–^* mice was counteracted by olmesartan through blockade of P38 phosphorylation.

### 2.4. Olmesartan Prevents Apoptosis and Inflammation in the Kidneys of Col4a3^–/–^ Mice

To address the consequences following MAPK activation that may accelerate kidney fibrosis, we investigated the alterations of apoptosis and inflammation in the kidneys of *Col4a3^–/–^* mice. Immunoblotting of whole kidney lysate, as seen in Figure 5a, revealed enhanced cleavage of caspase 3 and increased BAX/Bcl-2 ratio in *Col4a3^–/–^* mice. Morphologically, terminal deoxynucleotidyl transferase dUTP nick end labeling (TUNEL)-positive tubular epithelial cells in renal cortices were considerably increased in *Col4a3^–/–^* mice, as seen in Figure 5b, collectively suggesting that the apoptotic process is activated in the kidney, especially in the cortical tubular epithelial cells, of *Col4a3^–/–^* mice in comparison with WT mice. In addition, immunohistochemical staining of F4/80, a marker of macrophages, showed marked infiltration of macrophages in the interstitial space of *Col4a3^–/–^* mice, as seen in Figure 5b, implying concomitant inflammatory process. Indeed, qPCR using whole kidney lysate, as seen in Figure 5c, demonstrated significant upregulation of inflammatory cytokines and adhesion molecules, such as *Il-6*, *Tnfα*, *Vcam1*, and *Icam1*. Further, immunoblotting of whole kidney lysate, as seen in Figure 5d, showed upregulation of CD68 and hemoxygenase-1 in *Col4a3^–/–^* mice, further indicating underlying tissue inflammation. Notably, olmesartan treatment prevented such alterations in *Col4a3^–/–^* mice regarding apoptosis and inflammation, as seen in Figure 5a–d.

### 2.5. Olmesartan Targets RAS-TGFβ Feedback Loop to Regulate HK-2 Cell Fibrosis

To assess the hemodynamic benefits of olmesartan treatment on kidney protection in *Col4a3^–/–^* mice, we compared systolic and diastolic blood pressures (BPs) among the groups. In comparison with WT mice, both systolic and diastolic BPs were significantly elevated in *Col4a3^–/–^* mice. Surprisingly, although the mean values were decreased, olmesartan treatment in *Col4a3^–/–^* mice did not lower either systolic or diastolic BP significantly, as seen in Appendix A. On the basis of the striking tubulointerstitial damage and fibrosis in in *Col4a3^–/–^* mice following RAS and *TGFβ* signaling activation as well as the potent kidney protection by olmesartan with minimal benefits on BP, we hypothesized that a positive feedback loop may exist between RAS and *TGFβ* signaling pathway in tubulointerstitial compartments, which could be efficiently intervened by olmesartan. Following the observation of intense expression of TGFβ from the cortical epithelial cells of *Col4a3^–/–^* mice, as seen in Figure 2c, we conducted a series of in vitro experiments using a human proximal tubular epithelial cell line, HK-2 cell. To prove the putative positive feedback loop, we first treated recombinant human Ang II (rhAngII) in HK-2 cells, as seen in Figure 6. Following rhAngII treatment, ACE, TACE, and AT1R were upregulated and ACE2 and AT2R were downregulated in HK-2. Among MAPKs, phosphorylation of JNK and P38, but not ERK, was enhanced by rhAngII treatment. Phosphorylation of SMAD2/3 and expression of SMAD4 were also significantly increased in rhAngII-treated HK-2 cells, indicating upregulation of TGFβ downstreams followed by RAS activation. Subsequently, we treated recombinant human TGFβ (rhTGFβ) in HK-2 cells, as seen in Figure 7, to mimic the paracrine effect of TGFβ secreted from adjacent tubular epithelial cells. Treatment of rhTGFβ in HK-2 cells activated its downstream as phosphorylation of SMAD2/3 and expression of SMAD4 and αSMA increased. Phosphorylation of JNK and P38, but not ERK, was also enhanced. Notably, activation of RAS was prominently observed in rhTGFβ-treated HK-2 cells, supported by the upregulation of TACE, AT1R, and the downregulation of ACE2, confirming the positive feedback loop between RAS and TGFβ signaling in HK-2 cells. Finally, targeting the feedback loop, olmesartan significantly reversed molecular alterations resulting from rhTGFβ treatment in HK-2 cells.

## 3. Discussion

In the present study, we discovered a mechanism by which olmesartan impedes the progression of kidney fibrosis in *Col4a3^–/–^* mice, an experimental model of AS. We reported preferential upregulation of TGFβ signal in tubular epithelial cells of *Col4a3^–/–^* mice, which was effectively attenuated by olmesartan treatment. In addition to RAS activation in the kidney of *Col4a3^–/–^* mice, we demonstrated the feedback loop between RAS and TGFβ signal in kidney epithelial cells, providing the mechanistic insight how blockage of angiotensin receptors could effectively ameliorate renal tubulointerstitial damages in AS.

Although the deleterious effect of RAS activation on the kidney fibrosis has been well established, in which TGFβ signals serve as downstream pathways, the effect of TGFβ signaling on the activation of RAS has not yet been described. Our results indicate compelling evidence that stimulation of kidney tubular epithelial cells with TGFβ triggers activation of classical RAS pathways via modulation of MAPKs. We assume that P38 might have a pivotal role in this feedback loop between RAS and TGFβ signaling because phosphorylation of P38 was consistently enhanced in *Col4a3^–/–^* mice; this effect was also seen by stimulation with Ang II or TGFβ in HK-2 cells and was inhibited by olmesartan co-treatment in vivo and in vitro despite the variable phosphorylation pattern of specific MAPKs according to in vivo and in vitro experiments and mode of stimulation. Provided the nature of positive feedback loops that amplify the signals, the concept of RAS-TGFβ feedback loop in tubular epithelial cells facilitates a better understanding as to how unopposed RAS activation results in the catastrophic tubulointerstitial fibrosis by activation of TGFβ and vice versa; it also emphasizes the role of ARBs in targeting these feedback loops.

We demonstrated the reversal of ACE2 expression level simultaneously with a blockade of angiotensin receptors in *Col4a3^–/–^* mice. Whereas our group previously reported the protective effect of ACE2 replacement in *Col4a3^–/–^* mice [16], the augmentation of circulating ACE2 was not able to delay the development of diabetic nephropathy [26,27]. Results suggest that, despite disease-specific therapeutic efficacy of ACE2 in AS, the site of action is restricted to the urinary space, and that the regulation of intrarenal RAS is critical for treatment of AS. Therefore, our findings on the modulation of ACE2 in tubular epithelial cells by olmesartan treatment further offers grounds for the use of ARBs in AS aimed at activated intrarenal RAS. On the other hand, it should be further evaluated whether direct infusion of Ang-(1-7) may prevent tubulointerstitial fibrosis in *Col4a3^–/–^* mice. Ang-(1-7) consists of only seven amino acids [10] and is able to freely filtrate across the glomerular barrier to reach the urinary space. In consideration of its therapeutic potential in obstructive uropathy [17], it would also be intriguing to test results of Ang-(1-7) replacement in *Col4a3^–/–^* mice.

In parallel with phosphorylation of P38 [28,29], we observed apoptosis and inflammation in the kidney of *Col4a3^–/–^* mice. As downstream processes of RAS activation, inhibition of apoptosis or inflammation may protect kidney from further damage, contributing to fibrosis in this murine model of AS; however, previous reports are inconsistent. Indeed, inhibition of caspases with protease inhibitor zVAD-fmk resulted in the loss of protective feedback on excessive reactive oxygen species formation and phospholipase A2 activation in TNFα-stimulated mice, increasing the lethality [30]; however, administration of the anti-apoptotic agents, insulin-like growth factor-1 and zVAD-fmk, prevented kidney tissue damage after ischemia–reperfusion injury [31]. Those conflicting reports on the overall effect of caspase inhibition indicate the diverse functions of caspases depending on the type of injury and stress. Similarly, studies targeting macrophage infiltration, a surrogate marker of tissue inflammation, reported contradictory results depending on the time of intervention after initial injury [32], predominantly a result of the heterogeneity of its subpopulations. Therefore, despite the improvement of apoptotic and inflammatory features in the kidneys of *Col4a3^–/–^* mice with olmesartan treatment, it remains unclear whether specific inhibition of apoptosis or inflammation can reverse tissue fibrosis in this murine model of AS due to their complex mode of action in response to stimuli.

We unexpectedly observed that, although glomeruli are the primary site of defect in AS, activation of TGFβ signals was not prominent in the glomeruli, and damaged tubular epithelial cells expressed TGFβ more intensely in *Col4a3^–/–^* mice. One possible explanation is the temporal and spatial gaps between TGFβ expression and fibrosis in glomeruli and tubules. Since GBM is the primary site of genetic event in AS, ultrastructural alterations begin from glomeruli [33]. Therefore, we speculated that TGFβ signaling may have been upregulated earlier in glomeruli than in tubules. Accordingly, our immunohistochemical observations suggest that fibrosis should be more advanced in glomeruli than in tubules, resulting in the paradoxical paucity of TGFβ proteins in the glomeruli of *Col4a3^–/–^* mice at the time of sacrifice due to the relative lack of cellular components. Whereas the specific function of TGFβ in glomerulosclerosis of *Col4a3^–/–^* mice remains to be further elucidated, the contribution of TGFβ to tubulointerstitial fibrosis in autocrine or paracrine manners is clear. In this regard, our findings are notable in that tubular epithelial cells still express TGFβ in the kidneys with glomerulosclerosis; the intervention with ARBs prevents further progression of tubulointerstitial fibrosis even in the late stage of disease by interrupting RAS-TGFβ feedback loop.

## 4. Materials and Methods

### 4.1. Experimental Animals and Protocols

The experimental protocol was approved by the Animal Care Regulations (ACR) Committee of Chonnam National University Medical School (CNU IACUC-H-2018-21, date of approval 20-05-2018). Wild-type and *Col4a3^–/–^* mice [5] on a congenic 129X1/SvJ background were purchased from the Jackson Laboratory (Bar Harbor, ME, USA), and only male mice were used in this study. Olmesartan treatment began at four weeks of age, continued for three weeks, where olmesartan medoxomil (Sigma-Aldrich, St. Louis, MO, USA) at a dose of 5 mg/kg/day dissolved in drinking water was administered; only vehicle was supplied for the other experimental groups. Mice were sacrificed at seven weeks of age.

### 4.2. Serum and urine NGAL measurement by ELISA

Serum was isolated by centrifuging blood samples at 2000× *g* for 5 min at room temperature (RT). Urine samples were centrifuged immediately after collection at 8000× *g* for 5 min. NGAL level were determined with a commercial ELISA kit (R&D Systems, Minneapolis, MN, USA), according to manufacturer’s instruction.

### 4.3. Histology and Immunohistochemistry

The kidney was placed into 10% neutral buffered formalin (Sigma-Aldrich) for fixation overnight at RT. After brief washing with PBS, the fixed kidney was paraffin-embedded, sectioned into 3-μm thickness, stained, and scanned. Hematoxylin and eosin, (PAS), and Masson’s trichrome stainings were done, as previously described [34]. Periodic acid–Schiff staining followed the manufacturer’s instruction (Abcam, Cambridge, MA, USA). For immunohistochemistry, deparaffinized tissue sections were antigen-retrieved by heating at 100°C for 15 min in citrate buffer, pH 6.0 (Sigma-Aldrich). Following incubation with blocking buffer (1% (*w*/*v*) bovine serum albumin dissolved in 0.3% (*v*/*v*) Triton-X100 in PBS (0.3% PBS-T)) for 1 h at RT, the section was incubated with primary antibodies diluted in blocking buffer overnight at 4°C. After brief wash with 0.3% PBS-T three times, sections were incubated with horse radish peroxidase-conjugated secondary antibodies diluted in blocking buffer for 1 h at RT. After brief wash with 0.3% PBS-T three times and with PBS once, sections were incubated in 3,3′-diaminobenzidine reaction solution (Abcam). Following the rinse in tap water, the sections were counter-stained with Meyer’s hematoxylin solution, were washed in tap water, dehydrated in ethanol and xylene, and mounted. Primary and secondary antibodies used in immunohistochemistry are listed in Appendix A.

### 4.4. Detection of Apoptosis with Tunel Staining

Apoptosis of tubular epithelial cells was detected with TUNEL staining with ApopTag Plus Peroxidase In Situ Apoptosis Kit (Sigma-Aldrich), according to the manufacturer’s instruction.

### 4.5. Cell Culture

Human renal proximal tubular epithelial cells (HK-2 cells, American Type Culture Collection, Manassas, VA, USA) were cultured, as previously described. Briefly, cells were passaged approximately every three to four days in 100-mm dishes containing combined Dulbecco’s modified Eagle’s (DMEM) and Hams F-12 medium (Sigma-Aldrich) supplemented with 10% fetal bovine serum (FBS; Life Technologies; Gaithersburg, MD, USA), 100 U/mL penicillin, and 100 mg/mL streptomycin (Sigma-Aldrich). The cells were then incubated in a humidified atmosphere of 5% CO_2_ and 95% air at 37 °C for 24 h, and sub-cultured until 70–80% confluence. For treatment with recombinant proteins or chemicals, cells were plated onto 60-mm dishes in a medium containing 10% FBS and incubated for 24 h. The cells were then incubated in DMEM-F12 medium with 2% FBS and treated with rhAngII (1 μg/mL; Bachem, Bubendorf, Switzerlandsource) or rhTGFβ (2 ng/mL; R&D Systems) for an additional 16 h. Olmesartan medoxomil (5 ng/mL) was added 1 h prior to rhTGFβ treatment.

### 4.6. Semi-quantitative Immunoblotting

The kidney tissue was homogenized in modified RIPA buffer (150 mM sodium chloride, 50 mM Tris-HCl (pH 7.4), 1 mM EDTA, 1% *v*/*v* Triton-X 100, 1% *w*/*v* sodium deoxycholic acid, 0.1% *v*/*v* SDS). Cells were harvested by scrapping the plate in modified RIPA buffer. The tissue homogenates or cell lysates were centrifuged at 4000 g for 15 min at 4 °C. Total protein concentration in the supernatants was measured by bicinchoninic acid assay kit (Pierce; Rockford, IL, USA). All samples were adjusted to reach the same final protein concentrations. Then, the supernatants were boiled with loading buffer for 5 min. Proteins in tissue lysates were separated by 6−12% SDS-PAGE gel and were electrophoretically transferred onto nitrocellulose membranes (Hybond ECL RPN3032D; Amersham Pharmacia Biotech, Little Chalfont, UK) using Bio- Rad Mini Protean II apparatus (Bio-Rad, Hercules, CA, USA). After incubation with blocking buffer (80 mM Na_2_HPO_4_, 20 mM NaH_2_PO_4_, 100 mM NaCl, and 0.1% Tween-20 at pH 7.5) for 1 h at RT, the membranes were incubated with primary antibodies overnight at 4 °C. Following brief washing, the membranes were then incubated with primary antibodies for 1 h at RT. The immunoblots were detected with an enhanced chemiluminescence system kit (EMD Millipore). Densitometry was calculated by Scion Image software (Scion Corp, Frederick, MD, USA). Primary and secondary antibodies used in immunoblottings are listed in Appendix A.

### 4.7. Real-Time qPCR

Kidneys were homogenized in Trizol reagent (Invitrogen, Carlsbad, CA, USA). RNA was extracted from snap-frozen kidney tissue using the RNeasy Mini kit (Qiagen, Mississauga, Canada). cDNA was reverse-transcribed from 5 μg of total RNA with SuperScript II Reverse Transcriptase (Invitrogen). The relative level of tissue mRNA was determined by real-time qPCR, using a Rotor-GeneTM 3000 Detector System (Corbette research, Mortlake, New South Wales, Australia). The primers used in real-time qPCR are listed in Appendix A.

### 4.8. Measurement of BP

At the age of seven weeks, systolic and diastolic BPs were noninvasively measured from mouse tail using CODA Monitor (Kent Scientific, Torrington, CT, USA).

### 4.9. Statistical Analyses

Unless specified otherwise, results are expressed as mean ± SE. Two-tailed Student’s t-tests were used for comparisons between the two groups. One-way ANOVA with Newman–Keuls multiple comparison test was used for comparison of three groups or more. All statistical analyses were performed with the GraphPad Prism software (Version 5, GraphPad Software, La Jolla, CA, USA). A *p* value of less than 0.05 was considered significant.

## Figures and Tables

**Figure 1 ijms-20-03843-f001:**
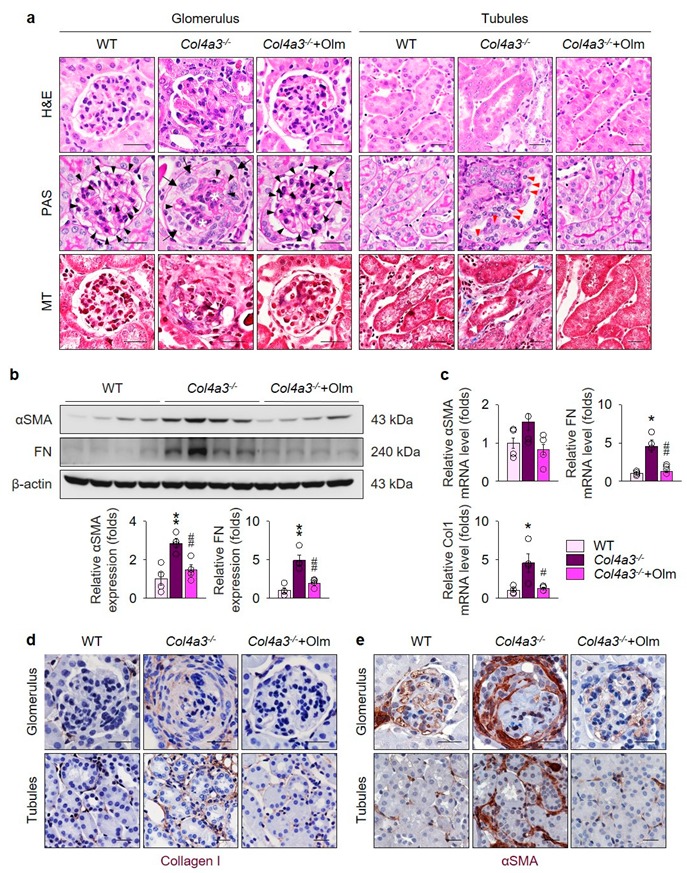
Olmesartan ameliorates kidney fibrosis in *Col4a3^–/–^* mice. (**a**) Tissue morphology of kidney from WT, *Col4a3^–/–^*, and *Col4a3^-^*^/-^+Olm mice. Images from glomerulus (left) and tubulointerstitium (right) are presented. H&E, hematoxylin, and eosin staining; PAS, periodic acid–Schiff staining; MT, Masson’s trichrome staining. Note that, contrary to kidneys from the other groups, glomerular capillary lumens are barely patent (arrowheads) with shrinkage of mesangium (white asterisk) in *Col4a3^–/–^* mice. In *Col4a3^–/–^* mice, crescent are packing the Bowman’s space (arrows). Detachment of tubular epithelial cells from basement membrane (red arrowheads) and collagen deposits (white arrows) are also indicated. Scale bars, 50 μm. (**b**,**c**) Comparison of expression level for fibrosis markers determined by immunoblotting (**b**) and qPCR (**c**) from the kidney of WT, *Col4a3^–/–^*, and *Col4a3^-/-^*+Olm mice (*n* = 4 mice/group). β-actin was used as the endogenous control. (**d**,**e**) Representative images of immunohistochemical staining for collagen type 1 (**d**) and αSMA (**d**) in the kidney of WT, *Col4a3^–/–^*, and *Col4a3^-/-^*+Olm mice. Images from glomeruli (upper) and tubules (lower) are presented. Scale bars, 50 μm. * *P* < 0.05, ** *P* < 0.01 versus WT mice; ^#^
*P* < 0.05, ^##^
*P* < 0.01 vs. *Col4a3^–/–^* mice by one-way ANOVA with Newman–Keuls multiple comparison test.

**Figure 2 ijms-20-03843-f002:**
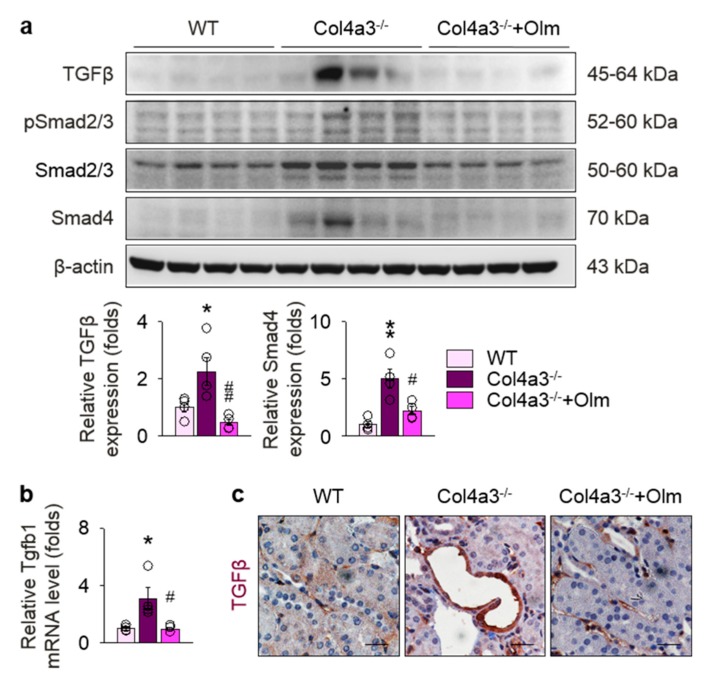
Upregulation of TGFβ and activation of its downstreams in *Col4a3^–/–^* mice is attenuated by olmesartan. (**a**) Comparison of protein expression level TGFβ and its downstream determined by immunoblotting from the kidney of WT, *Col4a3^–/–^*, and *Col4a3^-/-^*+Olm mice (*n* = 4 mice/group). β-actin was used as the endogenous control. (**b**) Comparison of mRNA expression-level TGFβ1 determined by qPCR from the kidney WT, *Col4a3^–/–^*, and *Col4a3^-/-^*+Olm mice (*n* = 4 mice/group). β-actin was used as the endogenous control. (**c**) Representative images of immunohistochemical staining for TGFβ in the renal cortex of WT, *Col4a3^–/–^*, and *Col4a3^-/-^*+Olm mice. Scale bars, 50 μm. * *P* < 0.05, ** *P* < 0.01 versus WT mice; ^#^
*P* < 0.05, ^##^
*P* < 0.01 versus *Col4a3^–/–^* mice by one-way ANOVA with Newman–Keuls multiple comparison test.

**Figure 3 ijms-20-03843-f003:**
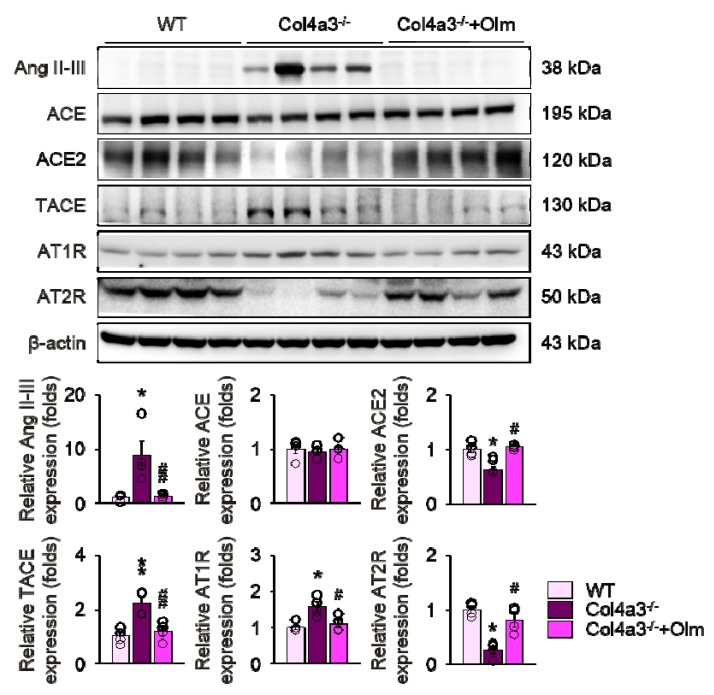
RAS activation in *Col4a3^–/–^* mice is counteracted by olmesartan. Comparison of protein expression level of RAS components determined by immunoblotting from the kidney of WT, *Col4a3^–/–^*, and *Col4a3^-/-^*+Olm mice (*n* = 4 mice/group). β-actin was used as the endogenous control. * *P* < 0.05, ** *P* < 0.01 versus WT mice; ^#^
*P* < 0.05, ^##^
*P* < 0.01 vs. *Col4a3^–/–^* mice by one-way ANOVA with Newman–Keuls multiple comparison test.

**Figure 4 ijms-20-03843-f004:**
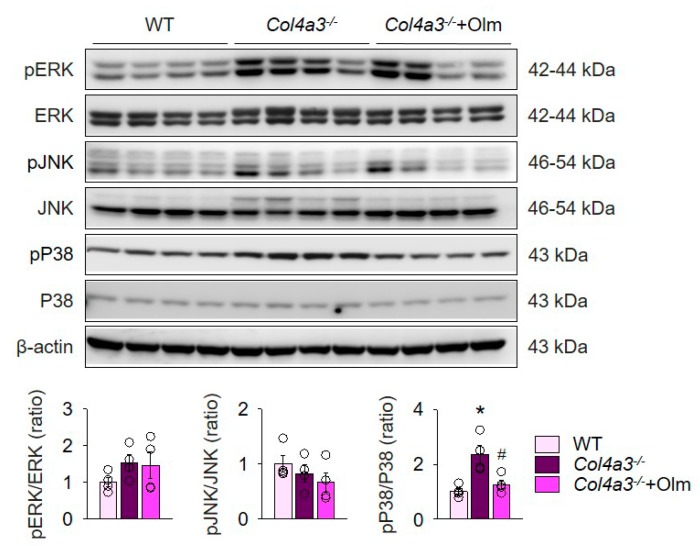
Olmesartan attenuates enhanced phosphorylation of P38 in *Col4a3^–/–^* mice. Comparison of protein expression level of MAPKs determined by immunoblotting from the kidney of WT, *Col4a3^–/–^*, and *Col4a3^-/-^*+Olm mice (*n* = 4 mice/group). β-actin was used as the endogenous control. * *P* < 0.05 versus WT mice; ^#^
*P* < 0.05 versus *Col4a3^–/–^* mice by one-way ANOVA with Newman–Keuls multiple comparison test.

**Figure 5 ijms-20-03843-f005:**
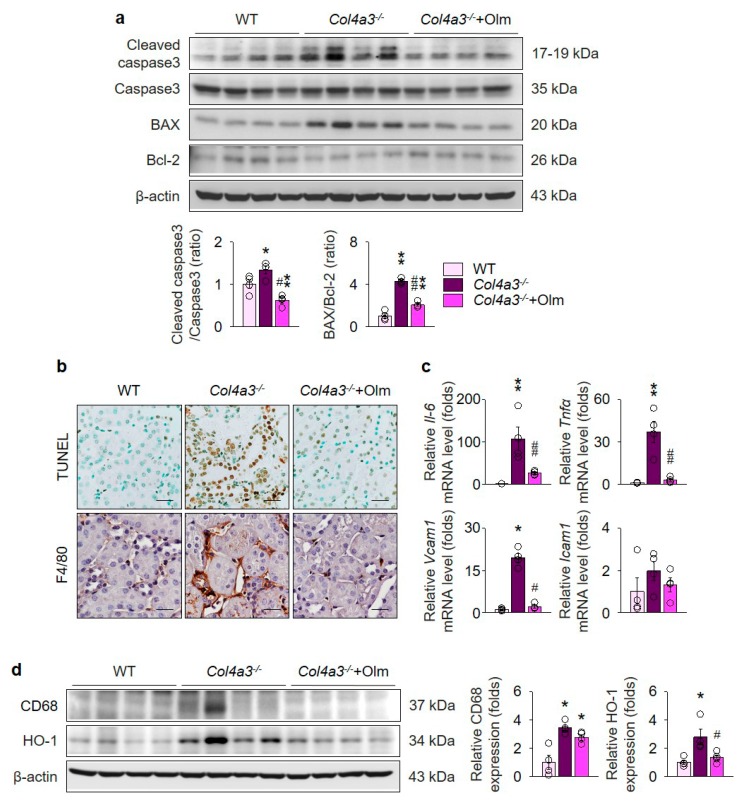
Olmesartan suppresses apoptosis and inflammation in the kidney of *Col4a3^–/–^* mice. (**a**) Comparison of protein expression level for molecules related to apoptosis determined by immunoblotting from the kidney of WT, *Col4a3^–/–^*, and *Col4a3^-/-^*+Olm mice (*n* = 4 mice/group). β-actin was used as the endogenous control. (**b**) Representative images of TUNEL (upper) and immunohistochemical staining for F4/80 (lower) in the kidney of WT, *Col4a3^–/–^*, and *Col4a3^-/-^*+Olm mice. (**c**) Comparison of mRNA expression level for inflammatory markers determined by qPRC from the kidney of WT, *Col4a3^–/–^*, and *Col4a3^-/-^*+Olm mice (*n* = 4 mice/group). β-actin was used as the endogenous control. (**d**) Comparison of protein expression level for CD68 and hemoxygenase-1 (HO-1) determined by immunoblotting from the kidney of WT, *Col4a3^–/–^*, and *Col4a3^-/-^*+Olm mice (*n* = 4 mice/group). β-actin was used as the endogenous control. Scale bars, 50 μm. * *P* < 0.05, ** *P* < 0.01 versus WT mice; ^#^
*P* < 0.05, ^##^
*P* < 0.01 versus *Col4a3^–/–^* mice by one-way ANOVA with Newman–Keuls multiple comparison test.

**Figure 6 ijms-20-03843-f006:**
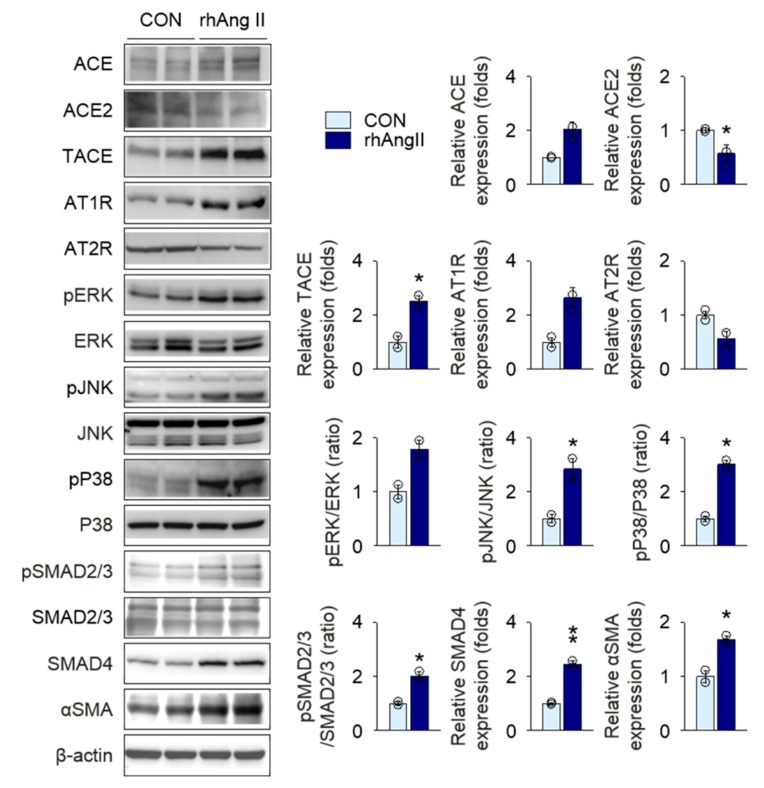
Stimulation with Ang II activates downstreams of TGFβ in HK-2 cells. Comparison of protein expression level for RAS components, MAPKs, and downstreams of TGFβ in HK-2 after stimulation with vehicles or recombinant human Ang II (rhAng II) (*n* = 3−4/group). Data are representative more than three independent experiments. * *P* < 0.05, ** *P* < 0.01 versus vehicle group by Student’s *T*-test.

**Figure 7 ijms-20-03843-f007:**
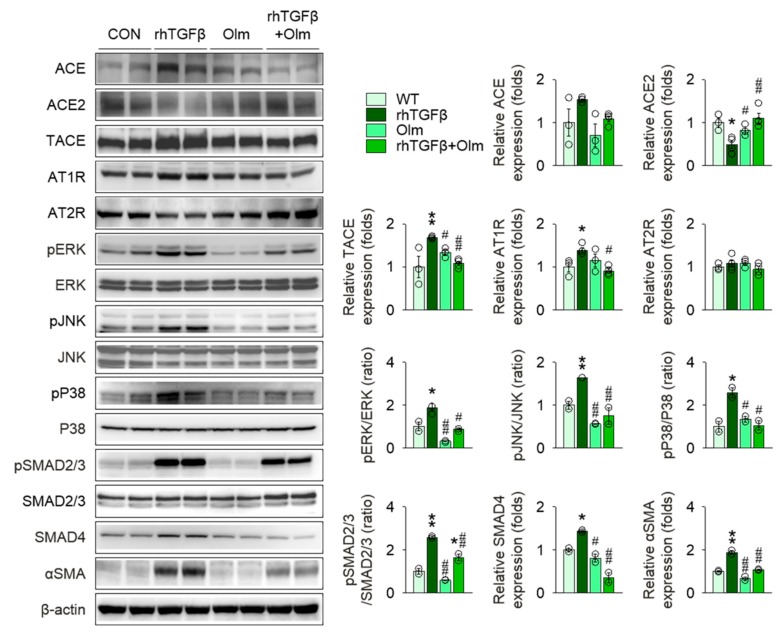
Olmesartan alleviates RAS activation in HK-2 cells stimulated with TGFβ. Comparison of protein expression level for RAS components, MAPKs, and downstreams of TGFβ in HK-2 cells. Cells were treated with vehicles, recombinant human TGFβ (rhTGFβ), Olmesartan medoxomil (Olm), or combination of rhTGFβ and Olm. (*n* = 3−4/group). Data are representative of more than three independent experiments. * *P* < 0.05, ** *P* < 0.01 versus vehicle group; ^#^
*P* < 0.05, ^##^
*P* < 0.01 versus rhTGFβ group by one-way ANOVA with Newman–Keuls multiple comparison test.

**Table 1 ijms-20-03843-t001:** Effect of Olmesartan on the renal function in *Col4a3^–/–^* mice.

	WT	*Col4a3^–/–^*	*Col4a3^-^*^/-^+Olm
Body weight (g)	19.7 ± 2.6	20.0 ± 0.4	20.7 ± 1.0
Kidney weight (g)	0.1 ± 0.0	0.1 ± 0.0	0.1 ± 0.0
Kidney weight / body weight (g/kg)	7.6 ± 1.6	9.3 ± 0.3	8.3 ± 0.4
Urine creatinine (mg/dL)	49.0 ± 10.5	21.2 ± 3.5 ^*^	9.1 ± 3.3 ^##^
Urine albumin (μg/mL)	7.2 ± 2.0	234.5 ± 32.5 ^*^	72.0 ± 45.1 ^#^
Urine albumin-to-creatinine ratio	15.1 ± 4.4	1119.0 ± 188.5 **	426.3 ± 105.1 ^**,##^
Serum NGAL (ng/mL)	77.2 ± 7.5	335.5 ± 126.1 *	43.8 ± 6.4 ^#^
Urine NGAL (ng/mL)	61.1 ± 3.8	848.8 ± 12.7 **	27.8 ± 2.8 ^##^

Values are expressed as mean ± SE. NGAL, neutrophil gelatinase-associated lipocalin. * *P* < 0.05, ** *P* < 0.01 versus WT mice; ^#^
*P* < 0.05, ^##^
*P* < 0.01 versus *Col4a3^–/–^* mice by one-way ANOVA with Newman–Keuls multiple comparison test.

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
