# Peer review of "Olmesartan Attenuates Kidney Fibrosis in a Murine Model of Alport Syndrome by Suppressing Tubular Expression of TGFβ"

_ijms, 2019, doi:10.3390/ijms20153843_

Round 1

Reviewer 1 Report

The authors have address my concerns providing well-argumented explanations

Author Response

Detailed Point-by-Point Response to Reviewers’ Comments

We deeply appreciate the editor and reviewers for their positive and favorable responses. We have revised the manuscript to address the issues raised by the reviewers.

Reviewer #1:

The authors have address my concerns providing well-argumented explanations.

Response: We appreciate the review’s encouraging comment.

Reviewer 2 Report

the paper is well designed and performed on a sound animal model frequently used for these experiments.

The data reported are interesting and are an added value to disclose the mechanism of action of olmesartan

the only question is to know about side effects of the drug treatment on these animals: did the authors observed any problem ? how many animals died during the treatment ?

Author Response

Detailed Point-by-Point Response to Reviewers’ Comments

We deeply appreciate the editor and reviewers for their positive and favorable responses. We have revised the manuscript to address the issues raised by the reviewers.

Reviewer #2:

The paper is well designed and performed on a sound animal model frequently used for these experiments. The data reported are interesting and are an added value to disclose the mechanism of action of olmesartan. The only question is to know about side effects of the drug treatment on these animals: did the authors observed any problems? How many animals died during the treatment?

Response: We appreciate the reviewer’s invaluable question. The use of ARBs, including olmesartan, in clinical practice is sometimes complicated with hyperkalemia and elevated serum level of creatinine, which are not measured in the present study. Instead, the gross features of the mice did not significantly differ, as the body and kidney weights are not substantially altered between Col4a3–/– and Col4a3-/-+Olm groups. We also assume that the chance of severe hyperkalemia should be very low with olmesartan administration in the current study, since the severe hyperkalemia leads to sudden cardiac arrest, but we did not observed any unexpected, premature loss of the mice from Col4a3-/-+Olm group. Therefore, the dose of olmesartan for oral administration (5 mg/kg/day) seems well-tolerated in mice, without significant adverse effects. We included a part of this explanation and rephrased the text in the revised manuscript.

Main text p.2: Compared to WT mice, urine creatinine and albumin concentration was significantly decreased and increased, respectively, in Col4a3–/– mice, despite the absence of significant differences in the body and kidney weight, and kidney-to-body weight ratio between the groups, implying the well-tolerated oral dosage of olmesartan in Col4a3–/– mice (Table 1).

This manuscript is a resubmission of an earlier submission. The following is a list of the peer review reports and author responses from that submission.

Round 1

Reviewer 1 Report

Suh et al. demonstrated that angiotensin receptor blocker (ARB), olmesartan attenuated kidney fibrosis via reducing transforming growth factor β (TGF-β) expression using Col4a3 22 –/– mice. The manuscript is well written and the findings are interesting. However, there are so many papers about the effect of angiotensin II on tissue fibrosis via TGF-β signaling. Moreover,treatment with ARB shows a beneficial effect on Alport syndrome in several papers. Therefore, there is less novelty and originality in this paper.

Author Response

Dear Sir/Madam,

We deeply appreciate the editor and reviewers for their thoughtful, critical and constructive comments, which have undoubtedly provided us with valuable opportunities to improve our work. We have rechecked experimental designs and revised the manuscript to address the issues raised by the reviewers.

Very sincerely yours, 

Eun Hui Bae & Soo Wan Kim

Detailed Point-by-Point Response to Reviewers’ Comments

We deeply appreciate the editor and reviewers for their thoughtful, critical and constructive comments, which have undoubtedly provided us with valuable opportunities to improve our work. We have rechecked experimental designs and revised the manuscript to address the issues raised by the reviewers.

Reviewer #1:

Suh et al. demonstrated that angiotensin receptor blocker (ARB), olmesartan attenuated kidney fibrosis via reducing transforming growth factor β (TGF-β) expression using Col4a3–/– mice. The manuscript is well written and the findings are interesting. However, there are so many papers about the effect of angiotensin II on tissue fibrosis via TGFβ signaling. Moreover, treatment with ARB shows a beneficial effect on Alport syndrome (AS) in several papers. Therefore, there is less novelty and originality in this paper.

Response: We appreciate the review’s fundamental comment. The authors agree that many other papers already demonstrated pro-fibrosis effect of angiotensin II via TGFβ signaling and beneficial effect of ARBs on Alport syndromes. We are, however, focusing on the detailed mechanism how olmesartan attenuates kidney fibrosis in Col4a3–/– mice. In this respect, our study is reporting several intriguing findings that have not been previously defined. We revealed preferential up-regulation of TGFβ signal in tubular epithelial cells of Col4a3–/– mice at a specific time point of sacrifice, which was effectively attenuated by olmesartan treatment. In addition to previously well-documented renin-angiotensin system (RAS) activation in the kidney of Col4a3–/– mice, we unveiled the feedback loop between RAS and TGFβ signal in kidney epithelial cells, providing the mechanistic insight how blockage of angiotensin receptors could effectively ameliorate renal tubulointerstitial damages in AS. These collectively present a biologic evidence that ARB treatment could be still effective even in a moderately advanced stage of AS, on top of previous reports that mainly emphasize early administration of RAS blockers (Gross et al, Kidney Int, 2003; Gross et al, Nephrol Dial Transplant, 2004). Therefore, the authors assume that our results deserve to be shared with fellow researchers to expand our understandings on the pathophysiology and treatment of Alport syndrome. We hope the reviewer favorably consider this context.

Reviewer 2 Report

The manuscript deals with the use of olmesartan in kidney fibrosis occurring in Alport syndrome. The authors use a well established mouse model of the disease lacking collagen 4alpha 3 gene. They demonstrate that olmesartan ameliorates kidney fibrosis supressing TGF-beta pahtway only in epithelial tubular cells, but not in glomeruli, which is rather surprising. In fact, TGFbeta immunstaining in AS mice is clearly downregulated in comparison wityh wild-type and treated mice. I disagree with the authors' sentence they textually say "...collectively, our findings suggest TGFbeta signaling is up-regulated more preferentially in tubular epithelial cells than in glomerulus... page5/line15-17). It is clear that tubular cells and glomerulus ones has clear contrast results, which deserves some more explanation than the simple last paragraph of the discussion. In fac,t olmesartan normalizes TGFbeta stainings in both epithelial tubular cells and glomeruli comparing with controls. Which does not fit well is the immunohistochemical results with western blot ones since the overall signal for TGFbeta in Col4a3-/- in kidney lisates is surprisingly high taking into account the contrast immunostainings occurring in both histological structures. I would expect that the final WB signal was lower since the increase in tubular cells might be neutralized by the decreased one of glomerulis. I would suggest to perform independent WB from glomeruli and tubular isolated fractions.

On the other hand, there is a mistake in figure 3, which is the same that figure 4. The former (RAS activation) is missing.

I would suggest to carry out some TEM  (transmission electron microscopy) to better illustrate the most important structural changes(fig.1a), which will help to identify well the impaired and normalized (after olmesartan treatment) histological structures.  

Author Response

Dear Sir/Madam,

We deeply appreciate the editor and reviewers for their thoughtful, critical and constructive comments, which have undoubtedly provided us with valuable opportunities to improve our work. We have rechecked experimental designs and revised the manuscript to address the issues raised by the reviewers.

Best regards,

Eun Hui Bae & Soo Wan Kim

Detailed Point-by-Point Response to Reviewers’ Comments

We deeply appreciate the editor and reviewers for their thoughtful, critical and constructive comments, which have undoubtedly provided us with valuable opportunities to improve our work. We have rechecked experimental designs and revised the manuscript to address the issues raised by the reviewers.

Reviewer #2:

The manuscript deals with the use of olmesartan in kidney fibrosis occurring in Alport syndrome (AS). The authors use a well-established mouse model of the disease lacking collagen 4 alpha 3 gene.

Comment 1: They demonstrate that olmesartan ameliorates kidney fibrosis suppressing TGFβ pathway only in epithelial tubular cells, but not in glomeruli, which is rather surprising. In fact, TGFβ immunostaining in AS mice is clearly downregulated in comparison with wild-type and treated mice. I disagree with the authors' sentence they textually say "...collectively, our findings suggest TGFβ signaling is up-regulated more preferentially in tubular epithelial cells than in glomerulus... page5/line15-17). It is clear that tubular cells and glomerulus ones has clear contrast results, which deserves some more explanation than the simple last paragraph of the discussion. In fact olmesartan normalizes TGFβ stainings in both epithelial tubular cells and glomeruli comparing with controls. Which does not fit well is the immunohistochemical results with western blot ones since the overall signal for TGFβ in Col4a3-/- in kidney lysates is surprisingly high taking into account the contrast immunostainings occurring in both histological structures. I would expect that the final WB signal was lower since the increase in tubular cells might be neutralized by the decreased one of glomeruli. I would suggest to perform independent WB from glomeruli and tubular isolated fractions.

Response: We appreciate this constructive comment. As the authors also consider the differential expression of TGFβ in glomeruli and tubular epithelial cells of Col4a3–/– mice as one of key findings in the present study, further discussions on this finding have already been included at the end of manuscript. We attribute this to temporal and spatial gaps between TGFβ expression and fibrosis in glomeruli and tubules. Since the glomerular basement membrane (GBM) is the primary site of genetic event in AS, ultrastructural alterations begin from glomeruli (Fogo et al, Am J Kid Dis, 2016). Therefore, we speculated that TGFβ signaling may have been up-regulated earlier in glomeruli than in tubules. Accordingly, our immunohistochemical observations was interpreted that fibrosis should be more advanced in glomeruli than in tubules, resulting in the paradoxical paucity of TGFβ proteins in the glomeruli of Col4a3–/– mice at the time of sacrifice due to the relative lack of cellular components. In this regard, it does not seem essential that immunoblotting of TGFβ for isolated glomerular and extra-glomerular factions to more clarify the up- or down-regulation of TGFβ in each tissue components, since our observation from immunoblotting and immunohistochemistry demonstrating up-regulation of TGFβ signal in 7-week old Col4a3–/– mice is in solid scientific contexts (Fogo et al, Am J Kid Dis, 2016; Gross et al, Kidney Int, 2003). Moreover, our data is comparable to the previously reported one, where immunoblotting of whole kidney lysate revealed the up-regulation of TGFβ signal in Col4a3–/– mice, which was significantly blocked by early, long-term ramipril treatment. Therefore, the authors determined that additional sacrifice of mice to obtain isolated glomerular and extra-glomerular protein factions could be barely justified in respect of ethics for animal experiment. Instead, we included more focused and detailed descriptions and discussions in the revised manuscript.

Main text p.5: Collectively, our findings suggest TGFβ signaling is up-regulated more preferentially in tubular epithelial cells than in glomerulus of Col4a3–/– mice at the time of sacrifice, which is efficiently attenuated by olmesartan treatment.

Main text p.11: One possible explanation is the temporal and spatial gaps between TGFβ expression and fibrosis in glomeruli and tubules. Since GBM is the primary site of genetic event in AS, ultrastructural alterations begin from glomeruli (Fogo et al, Am J Kid Dis, 2016). Therefore, we speculated that TGFβ signaling may have been up-regulated earlier in glomeruli than in tubules. Accordingly, our immunohistochemical observations was interpreted that fibrosis should be more advanced in glomeruli than in tubules, resulting in the paradoxical paucity of TGFβ proteins in the glomeruli of Col4a3–/– mice at the time of sacrifice due to the relative lack of cellular components.

Comment 2: On the other hand, there is a mistake in figure 3, which is the same that figure 4. The former (RAS activation) is missing.

Response: We appreciate this critical comment and apologize for the mistake in the manuscript editing. We included an appropriated date in Figure 3.

Figure 3. RAS activation in Col4a3–/– mice is counteracted by olmesartan. Comparison of protein expression level of RAS components determined by immunoblotting from the kidney of WT, Col4a3–/–, and Col4a3-/-+Olm mice (n = 4 mice/group). β-actin was used as the endogenous control. *P < 0.05, **P < 0.01 vs. WT mice; #P < 0.05, ##P < 0.01 vs. Col4a3–/– mice by one-way ANOVA with Newman-Keuls multiple comparison test.

Comment 3: I would suggest to carry out some TEM (transmission electron microscopy) to better illustrate the most important structural changes(fig.1a), which will help to identify well the impaired and normalized (after olmesartan treatment) histological structures.

Response: We appreciate this constructive comment. The authors initially planned electron microscopic study to elucidate the morphological defects of Col4a3–/– mouse kidney and the mode of act of olmesartan at ultrastructural levels. The plan was, however, soon discouraged, as previous studies already described ultrastructural findings of Col4a3–/– mouse kidney treated with or without ACEi/ARBs (Gross et al, Kidney Int, 2003; Gross et al, Nephrol Dial Transplant, 2004). Although the characteristic thickening and splitting of GBM was observed both in treated and untreated Col4a3–/– mice, treatment with ramipril slightly improved foot process effacement, glomerular hypertrophy, and accumulation of intracellular fibrillar collagens. Those studies demonstrated an outstanding anti-fibrotic effect of ACEi or ARBs in the glomeruli and tubulointerstitium of Col4a3–/– mice. We included a part of this explanation and rephrased the text in the revised manuscript.

Main text p.2: Indeed, pre-emptive treatment with an angiotensin-converting enzyme inhibitor (ACEi) or angiotensin II receptor blocker (ARB), despite the persistent ultrastructural defect in GBM, suppressed cytokine and collagen production, and consequently attenuated inflammation and tubulointerstitial fibrosis in Col4a3–/– mice (Gross et al, Kidney Int, 2003; Gross et al, Nephrol Dial Transplant, 2004). Ultrastructural analyses in those studies demonstrated that treatment with ramipril slightly improved foot process effacement, glomerular hypertrophy, and accumulation of intracellular fibrillar collagens, although the characteristic thickening and splitting of GBM was observed both in treated and untreated Col4a3–/– mice. Those studies also reported an outstanding anti-fibrotic effect of ACEi or ARBs both in the glomeruli and tubulointerstitium of Col4a3–/– mice.